# Self-Refereeing System in Ultimate during the Joint Junior Ultimate Championship in Three Different Divisions—A Different Way to Promote Fair-Play?

José Pedro Amoroso [1,2,*], Luís Coelho [1,2], Henrietta Papp [3], Felipe Costa [4], Efstathios Christodoulides [5], Wouter Cools [6,7], Zoltán Erdősi [8], James E. Moore, Jr. [9] and Guilherme Eustáquio Furtado [10,11,*]

1   ESECS—Polytechnic of Leiria, 2411-901 Leiria, Portugal; coelho@ipleiria.pt
2   CIEQV—Life Quality Research Center, Polytechnic of Leiria, 2411 Leiria, Portugal
3   National Laboratory of Virology, Szentagothai Research Centre, University of Pécs, 7624 Pécs, Hungary; phencsi@gmail.com
4   Faculty of Physical Education, Brasilia University, UnB, Brasília 70910-900, Brazil
5   Sport and Exercise Sciences, School of Sciences, UCLan Cyprus, 7080 Larnaka, Cyprus
6   Multidisciplinary Institute of Teacher Training (MILO), Vrije Universiteit Brussel, 1050 Brussels, Belgium
7   Department of teacher training for secondary education (OSO), Artevelde University of Applied Sciences, 9000 Ghent, Belgium
8   School of Doctoral Studies, Hungarian University of Sports Science, Alkotás u. 42-48, 1123 Budapest, Hungary
9   Department of Bioengineering, Imperial College London, London SW7 2AZ, UK
10  Applied Research Institute, Polytechnic Institute of Coimbra, Rua da Misericórdia, Lagar dos Cortiços–S. Martinho do Bispo, 3045-093 Coimbra, Portugal
11  Research collaborator Unit for Sport and Physical Activity (CIDAF, UID/PTD/04213/2020), Faculty of Sport Sciences and Physical Education (FCDEF-UC), Pavilhão 3, 3040-248 Coimbra, Portugal
*   Correspondence: jose.amoroso@ipleiria.pt (J.P.A.); guilherme.furtado@ipc.pt (G.E.F.)

**Abstract:** In ultimate games governed by the World Flying Disc Federation (WFDF), all competitors also take on the role of referee. The players discuss disputed calls with each other during the game, and then follow rules designed for these situations to determine how the play continues. The number one rule of the sport is to respect the spirit of the game (SOTG), which encourages competitive play while preserving mutual respect and minimizing the risk of injury. The use of SOTG in ultimate in the framework of self-arbitration as a moral practice aligns well with other tools of critical pedagogy. For this study, the SOTG scores of the WFDF Joint Junior Ultimate Championship (JJUC 2022) were analyzed. A total of 1009 players from 19 countries competed in 434 self-refereed games (29 national teams in the WJUC Under-20 (U20) tournament and 20 teams in the EYUC Under-17 (U17) tournament). All the scores from the individual criteria correlated well with the overall scores, but for the most part, they did not correlate with each other. Our experience with the scoring system has highlighted the importance of participants understanding the meaning of the results and how they may lead to a constructive reflection to improve exceptions, including scores representing fouls and rules knowledge. The findings provide important information for physical education teachers, coaches, and sport consultants and may be of use to design SOTG programs that could foster the experience of sportsmanship and to facilitate the ethical conduct of athletes in either recreative or in competitive contexts.

**Keywords:** self-regulation; teamwork; youth; competition; sportsmanship; fair play; sport pedagogy; psychology of sports; self-arbitration

## 1. Introduction

Self-refereed sports have the potential to encourage development of soft skills for every participant. Each game creates its own microcosmic world with its own standards of excellence and its own ways of failing. Context is provided by sports rules, and there

is always an expectation for the players to respect them [1]. We assume that ultimate has additional power as a self-arbitrated sport with the capacity of differentiating and adding sport-rich valences such as implying the assumption of error and accepting the decisions of others [2]. These are traits that sports should foster, as referees are often passive in their pedagogical function and perhaps unwilling to engage in the complexity of their respective responsibilities [3,4].

The presence of referees may create an environment in which players might try to skirt the rules to gain an advantage. Self-refereeing, on the other hand, brings an implicit obligation for a higher degree of respect for the rules, and mutual resolution of infractions [1]. When students are encouraged to focus on self-referenced improvement, they are more likely to make favorable judgments and to develop a growth mindset [5]. According to Ade et al. (2018), self-refereeing provides an opportunity to share the responsibilities of refereeing among the players and alleviate the stresses of rules infractions [6]. There are also refereed sports that are moving towards incorporating self-respect and respect for the opponent and the game itself [7].

Sports has changed over the past 40 years, from a world set up almost exclusively by and for boys and men to one that is moving substantially (though incompletely) towards gender equity [4]. With the clear changes in the expansion of Olympic sports, we must seek solutions adjusted to this new reality, as the Olympic Games have had the strength to inspire generations [8]. In the scientific literature, the impact of participation in sports, such as ultimate, that use self-refereeing as a central tenet, even as the highest levels of competition, has rarely been studied [9]. On the other hand, in sports that have had the support of various technologies, such as in elite football, it seems likely that refereeing will continue to be a contentious issue [10].

Implementing a self-arbitration system requires careful thought and consideration of the context of all specific sports [9]. This process has been instilled in ultimate since its beginnings. Ultimate is a non-contact, challenging team sport [11,12]. Its characteristics, such as the game environment and specific rules, appear to influence the on-the-pitch behavior of players and could enrich the school context by providing an effective context for teaching problem-solving and critical inquiry [6,9]. A negotiated curriculum in which students are actively involved in making decisions about what they do and how they do it has been proposed by Bovill and Bulley [13]. This may support PE promoting soft skills during the development of healthy and physically active behaviors and game-related motor skills [11,14]. This may facilitate the establishment of a unique culture sustained by students.

The Declaration on Sport, Tolerance, and Fair Play agreed that sports is an important field of education [15,16]. Fair play should be considered not only in a sporting dimension, but should be viewed as a universal principle that can be applied to various contexts in which young people are involved in regular sports [16], as well as a part of everyday life, including all the desired behaviors of sportspersons in their social functioning [17]. In youth, there is no difference in relation to gender in the understanding of the concept of fair play in the context of participation in sports [18]. However, most young people perceive fair play only in the sphere of sports [15,16,18].

SOTG was established as one of the core elements when flying disc sports were founded. It encompasses fair play and sportsmanship, which are strongly emphasized in ultimate [19,20]. The integrity of the sport depends on each player's responsibility to uphold SOTG, and this responsibility should not be taken lightly [21]. Since ultimate is a self-refereed sport, maintaining the spirit of the game is essential [22]. Players must know the rules, be fair-minded and truthful, explain their viewpoint clearly and briefly, allow opponents a reasonable chance to speak and resolve disputes as quickly as possible, using respectful language [23].

Directly after a game, each team rates their opponent team through a Likert scale ranging from 0 to 4 with closed questions that measure the five fundamentals of the sport: 1. *Did they know and abide by the rules*? 2. *Did they avoid body contact*? 3. *Were they fair-minded*?

4. *Did they show self-control and a positive attitude*? 5. *Did they communicate properly and respectfully*? Spirit scoring is especially recommended for leagues and larger tournaments. In these events a team's spirit captain (SC) should be responsible for collecting spirit scores and giving them to the spirit director (SD) [19].

According to our experience, SOTG is generally well respected among the players. The SOTG scores not only help the opponent to behave better on the field, but the team who received the highest score in the tournament is also awarding a prize, which is as important and satisfying as being 1st, 2nd, or 3rd in the ranking of the tournament. Spirit circles (SCi) are a good way to positively connect with the other team and to resolve possible conflicts. After a game is over, both teams form a joined circle with alternating players. This circle can be used to highlight some positives and/or discuss issues that might have occurred during the game. The objectives of the SOTG scoring system are to: (i) educate players on what SOTG is; (ii) help teams to improve specific aspects of their spirit; and (iii) celebrate SOTG by awarding a prize to the team that earns the highest score [20].

As team sports evolve into the professional sphere, players become fitter, faster, stronger, and obtain a more advanced knowledge of the rules of the game [24]. With this constant evolution, it is important to study new possibilities that help referees to judge players capable of representing the best of sports [25]. We need to study how (self) referees can improve their capability to judge a fellow player's penitential behavior to promote and show the high values that SOTG has to offer. Specifically, teamwork was conceptualized as "a collaborative effort by team members to effectively carry out the independent, and interdependent behaviors that are required to maximize a team's likelihood of achieving its purposes" [26]. The pressure on sports officials to adjudicate perfectly is increasing and with media pundits attempting to create controversy, the spotlight is often cast upon match referees [27]. Another dilemma concerns the acceptance of referee decisions, and professional sports players indicated an attitude undermining the role of a referee [16]. On the other hand, in highly competitive matches, a teams' desire to win occasionally supersedes compliance to follow and abide by SOTG and other participation norms [28].

Youth sports programs typically transition from recreational to performance-based goals, and cultural shifts making parental moral worth dependent on the personal achievements of their children makes youth sport a key setting for positive development [29]. To effectively promote the adoption and maintenance of active lifestyles, physical activity (PA) professionals should consider intervening on multiple levels regarding the adolescents' perspectives and the specific needs of the subgroups (e.g., perceptions of body image, femininity, friends' influence, available PA programs, and safety) [30]. On the other hand, a deep and structured involvement with one activity may restrain diverse social interactions, which are key elements that ultimate promotes [31].

Positive sporting experiences may provide young people with a better opportunity to realize benefits stemming from social connections, along with a sense of relatedness, competence, and achievement [32]. This can direct PE teachers to consider their personal philosophy of teaching physical education in regards to what a student should learn and why [33]. Coaches and parents can create environments in which pro-social behavior (e.g., verbally encouraging a teammate) is encouraged and reinforced through positive messaging, with further encouragement through team building, pregame player introductions, close supervision of practices and games, postgame social events, rewards for good sportsmanship, and appropriate consequences for poor sportsmanship [34]. The TAFISA games differ from professional sports, as victory is not the priority. Indeed, what is important is that the games are traditional and accessible for all [35]. Youth sport leaders, including program administrators, coaches, and parents, can positively address almost all of the reasons given for sports dropout among youth [36].

The U17 division was sanctioned by EUF, while the U20 division was sanctioned by the WFDF. Each division had an SD assisted by two additional volunteers. Here, we studied the scores given by the youth teams at the JJUC. The purpose of this study is to examine the SOTG results at JJUC 2022 in different divisions through self-refereeing and

determining whether players of different competitive levels demonstrated similar SOTG results.

## 2. Materials and Methods

### 2.1. Study Design

This retrospective team sports study involved players attending the World Youth/Junior Ultimate Championship (WJUC) and the European Youth Ultimate Championship (EYUC), which were held together (JJUC) in Wroclaw, Poland, from 6 to 13 August 2022. Our cross-sectional study involves a total sample of 1009 players: 454 were female, 554 were male, and 1 was gender diverse. There were 398 players at EYUC, 228 male, 170 female, while at WJUC, there were 609 players, 1 gender diverse, 284 female, and 324 male.

### 2.2. Participants and Settings

Figure 1 presents the divisions, the number of national teams, and the number of games played. The present study comprised players from 19 countries (Austria, Belgium, France, Great Britain, Hungary, Canada, Colombia, Czech Republic, France, Germany, Israel, Italy, New Zealand, Netherlands, Poland, Slovakia, Switzerland, Sweden, and the United States). Overall, 49 teams competed, 29 national teams played in the WJUC (U20) tournament and 20 teams in the EYUC (U17) tournament. The teams played for championship titles in Open, Women's, and Mixed divisions in *the Pola Marsowe complex*.

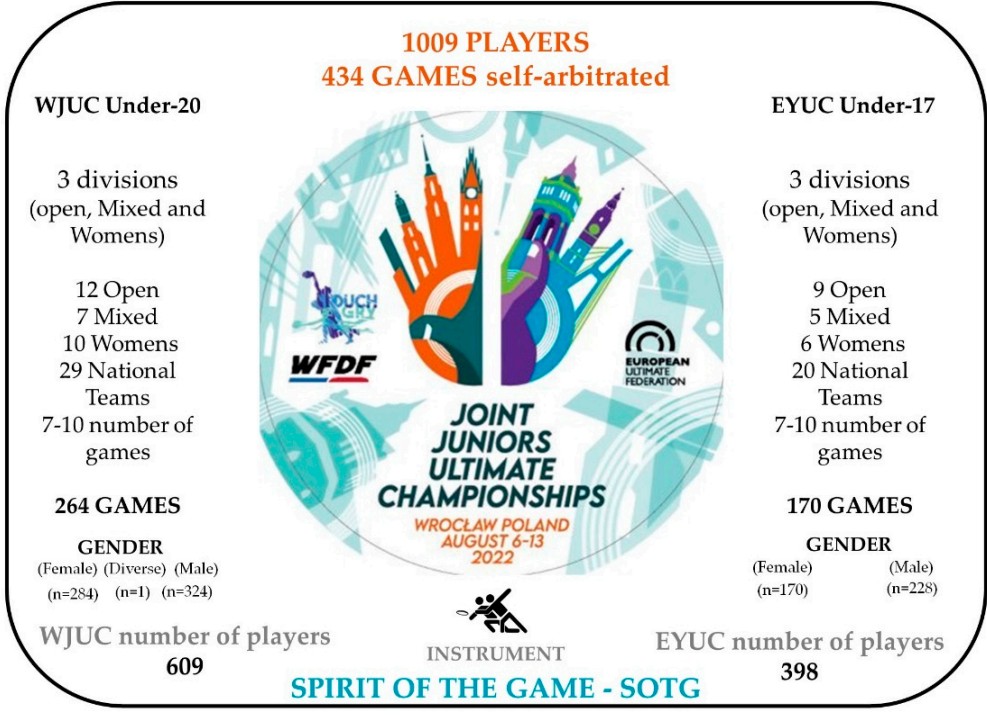

**Figure 1.** Joint Junior Ultimate Championship description, and their respective divisions: in 2022, the World and European Youth Championship were hold together in Wroclaw, Poland. A total of 434 self-refereed games were played with 1009 participating players altogether. Teams competed in three divisions, Open, Mixed, and Women's. After each game, the teams evaluated the opponent team's knowledge of the rules, body contact avoidance, attitude, communication, and fair-mindedness during their game. These are the main aspects of the spirit of the game.

Additionally, Figure 1 displays the relevant competition categories. The breakdown among the three divisions was: U17 open (Belgium, France, Great Britain, Germany, Israel, Italy, Netherlands, Switzerland, and Sweden, with a total of nine national teams); U17 women's (Belgium, France, Great Britain, Germany, Italy, and Sweden, with a total of six national teams); U17 mixed (Austria, Belgium, Hungary, Italy, and Poland, with a total

of five national teams); U20 open (Austria, Belgium, Canada, Colombia, Czech Republic, France, Great Britain, Germany, Italy, New Zealand, Poland, and the United States, with a total of twelve national teams); U20 women´s (Austria, Canada, Czech Republic, France, Great Britain, Germany, Italy, New Zealand, Poland, and the United States, with a total of ten national teams); U20 mixed (Colombia, Hungary, Israel, Netherlands, Switzerland, Slovakia, and Sweden, with a total of seven national teams).

### 2.3. Data Collection

After each game, the data was collected using the following procedures: (i) the SC facilitated an SCi with the opposing team. If for some reason, there was no time to set up an Sci, the SCs at least checked in with the opposing team's SC to share any quick thoughts and to decide if further discussion was needed; (ii) the teams evaluated their opponent teams promptly on the five principles of SOTG. A total of 434 self-refereed games were played. The whole of each team was required to be engaged in scoring, using it as an opportunity to reflect on the game and their own team's spirit; (iii) scores were entered or returned promptly to tournament organizers or scorekeepers; and (iv) all SOTG scores were saved into a digital spreadsheet. During the tournament, the spirit team, led by the spirit directors of the EYUC and WJUC, constantly monitored the scores. At the end of JJUC, all scores were locked and saved online.

### 2.4. Ethical Issues

Permission for data collection was sought from the World Flying Disc Federation followed the Declaration of Helsinki and produced by the World Medical Association for research with humans [37]. In this study, we did not use personal data, and only general anonymized data were analyzed.

### 2.5. Instrumentation

The SOTG scoring system was measured based on a marking system used immediately after each game [11]. The SOTG was measured by the sum of the scores obtained in five questions addressing the following domains: (1) Knowledge and use of the rules; (2) Fouls and body contact; (3) Fair-mindedness; (4) Positive attitude and self-control; (5) Communication. Answers were given on a 5-point Likert scale (0 = Poor; 1 = Not Good; 2 = Good; 3 = Very Good; 4 = Excellent). After each game, players rated whether the other team was "better than," "worse than," or "the same as" a rival in a regular game, using the anchor "Good" as a baseline for comparison. The final SOTG score is the sum scoring/marking and may vary between 0 and 20, where a score of 10 is considered normal, good SOTG [38].

### 2.6. Tournament Format

According to the WFDF [38]:

(i) In the EYUC (U17) Open Division, one pool of open teams ($n = 9$) played a round robin. The top 2 teams played the gold medal game. The 3rd and 4th place teams played the bronze medal game. The 5th through 8th place teams played placement games;

(ii) In the U17 Women's Division, one pool ($n = 6$) played a round robin. The top 3 teams formed Pool B, while the 4th, 5th and 6th place teams formed Pool C. No scores were carried forward from the first round. Following pool play, the top 2 teams from Pool B played the gold medal game. The 3rd place team from Pool B played the 1st place team from Pool C in the bronze medal game. The 2nd and 3rd place teams from Pool C played a placement game for 5th–6th place;

(iii) In the U17 Mixed Division tournament, mixed teams ($n = 5$), 2 round robins; the second round was determined two days after the first round, with U17 Open teams ($n = 9$) and U17 Women teams ($n = 6$), with round robins in one pool. The Upper and Lower power pool of 3 + round robins in those placement games earned 1st, 3rd and 5th placements;

(iv) In the WJUC (U20) Open Division two pools of open teams (*n* = 6 each) played a round robin. The top 2 teams from each pool formed Pool C, while the 3rd and 4th place teams from each pool formed Pool D, and the 5th and 6th place teams of each pool formed Pool E. All previous scores from the first round were carried forward to Pools D-E. The top 2 teams from Pool C moved on to participate in the semifinals. The 3rd and 4th place teams from Pool C and the 1st and 2nd place teams from Pool D played crossover games, with the winners of those games taking the remaining spots in the semifinals. The losers of the crossover games played a placement game for 5th–6th place. The 3rd and 4th place teams from Pool D and the 1st and 2nd place teams from Pool E moved to a 4-team placement bracket for 7th to 10th place. The 3rd and 4th place teams from Pool E played a placement game for 11th–12th place;

(v) In the U20 Women's Division, two pools of women's teams (*n* = 5) played a round robin. The top 2 teams from each pool formed Pool C. The 3rd, 4th, and 5th teams from each pool formed Pool D. In Pool C, all games were replayed. All previous scores from the first round were carried forward to Pool D. The top 2 teams from Pool C moved on to the semifinals. The 3rd and 4th place teams from Pool C and the 1st and 2nd place teams from Pool D played crossover games, with the winners of those games moving on to the semifinals. The losers of the crossover games played a placement game for 5th–6th place. The 3rd, 4th, 5th and 6th teams from Pool D moved to a 4-team placement bracket for 7th to 10th place;

(vi) In the U20 Mixed Division, one pool of teams (*n* = 7) played a round robin. The top 3 teams formed Pool B. The 4th, 5th, 6th and 7th place teams formed Pool C. All games were replayed; no scores were carried forward from the first round. The top 2 teams from Pool B played the gold medal game. The 3rd place team from Pool B and the 1st place team from Pool C played the bronze medal game. The 2nd, 3rd, and 4th place teams from Pool C did not play any more games.

### 2.6.1. Scoring of Spirit of the Game

Scoring SOTG is a team effort, and this value helps in educating new players and reinforces the fundamentals of SOTG with the more experienced players [11]. While scoring SOTG can take up to fifteen minutes the first few times, it will take only a few minutes after teams get used to it. The SOTG score is needed immediately after the game.

### 2.6.2. Guidelines for Spirit scores

Figure 2 shows the SOTG score sheet. The system was designed in accordance with the expectation that teams generally display normal, good spirit. Therefore, the baseline in each category is "Good" which equals 2 points. For each category, the team needs to determine together if the other team was better than, worse than, or the same as a rival in a regular game and score it accordingly. This is designed to produce a normal distribution of SOTG scores, with a mean of 10. This allows for the identification of outliers for the purposes of awarding the best spirited team and identifying teams that need to work on better SOTG.

During SOTG scoring, it is advised that teams discuss each category separately. Players are provided with an example to guide them on how to score the opposing team, allowing them to give a real score that reflects the items detailed on the sheet. There might be many times that nothing out of the ordinary occurred at a game. As such, each category should get 2 points. A final score of ten points is considered normal, good spirit. Teams should not give a team a higher score because of remarks made in the Spirit circle or any other post-game interactions. Rather, the scores should focus on the match itself; teams should not give lower Spirit scores out of retaliation or prejudice. If a team gives a zero or four, they must leave a comment to justify their extreme score, and to provide actionable feedback in the case of a low score. Comments for scores of one or three are recommended, and each team should always decide scores as a group (Figure 2).

**Figure 2.** SOTG game score sheet made up of five questions addressing the following domains: (1) Knowledge and use of the rules; (2) Fouls and body contact; (3) Fair-mindedness; (4) Positive attitude and self-control; (5) Communication. Answers were provided on a 5-point Likert scale (0 = Poor; 1 = Not Good; 2 = Good; 3 = Very Good; 4 = Excellent). After each game, players rated if the other team was "better than," "worse than," or "the same as" a rivals in a regular game, using the anchor "Good" as a baseline for comparison. The final SOTG score is the sum scoring/marking and may vary between 0 and 20, where a score of 10 is considered normal, good Spirit.

Rules of ultimate are recommended for all levels of play, from leagues to larger tournaments (WFDF Rules of Ultimate 2021–2024-Official Annotations-Official Version effective 2021-01-01, produced by the WFDF Ultimate Rules Sub-Committee. Rule 5.2. The team captain is a team member who is eligible to participate in the game and has been designated to represent the team in decision-making before, during, and after a game. The spirit captain discusses and resolve spirit issues at any point throughout the competition with opponents, teammates, coaches, and game or event officials).

## 3. Results

*Statistical Analysis*

Statistical calculations included several groups of analysis. First, a Kolmogorov–Smirnov test was used to check the normality of distribution. Means and standard deviations were calculated for all observed variables. Non-parametric inferential statistical models were used to compare SOTG overall scores and categories (Friedman and Wilcoxon) and divisions (Kruskal–Wallis H and Man–Whitney U), and the Spearman correlation was used to examine the existence of any type of association between all variables, with respect

to the following scale (Santos, 2022). For all the analyses, IBM SPSS Statistics for Windows, Version 28.0. (Armonk, NY, USA: IBM Corp.) was used, with a *p*-level of 95%.

With all criteria rated on a scale from 0 to 4, Positive attitude and self-control received the highest average score (2.44 ± 0.67), followed by Fair-mindedness (2.14 ± 0.56), Communication (2.08 ± 0.50), Fouls and body contact (1.86 ± 0.51), and Knowledge and use of the rules (1.79 ± 0.53), for the U17 competition (Table 1). Statistical differences were found between all SOTG domains ($p < 0.001$); nevertheless, "Fouls and body contact," "Knowledge and use of the rules," and "Communication and Fair-mindedness" showed to be statistically similar ($p \geq 0.05$). For the U20 division, Table 1 shows that Positive attitude and Self-control also showed the highest average score (2.36 ± 0.64), followed by Communication (2.25 ± 0.63), Fair-mindedness (2.15 ± 0.66), Fouls and body contact (1.90 ± 0.59), and Knowledge and use of the rules (1.84 ± 0.43). SOTG results between divisions showed no statistical differences, except for Communication, where U20 reveals higher scores ($p < 0.01$).

**Table 1.** Overall and detailed SOTG scores for U17 and U20 divisions.

| U17 | *n* | M | SD | Min. | Máx. | *p* |
|---|---|---|---|---|---|---|
| SOTG | 20 | **10.31** | 1.74 | 3 | 14 | - |
| Knowledge and use of the rules | 20 | **1.79** | 0.53 | 0 | 3 | |
| Fouls and body contact | 20 | **1.86** | 0.51 | 1 | 3 | |
| Fair-mindedness | 20 | **2.14** | 0.56 | 0 | 3 | <0.001 |
| Positive attitude and self-control | 20 | **2.44** | 0.67 | 1 | 4 | |
| Communication | 20 | **2.08** | 0.50 | 1 | 3 | |
| **U20** | | | | | | |
| SOTG | 29 | **10.50** | 1.91 | 3 | 16 | - |
| Knowledge and use of the rules | 29 | **1.84** | 0.43 | 0 | 3 | |
| Fouls and body contact | 29 | **1.90** | 0.59 | 0 | 3 | |
| Fair-mindedness | 29 | **2.15** | 0.66 | 0 | 4 | <0.001 |
| Positive attitude and self-control | 29 | **2.36** | 0.64 | 1 | 4 | |
| Communication | 29 | **2.25** | 0.63 | 0 | 4 | |

Notes: U17 = under 17; U20 = under 20; M = mean; *n* = sample size; SD = standard deviation; Min = minimum; Max. = maximum.

For the U20 division, Positive attitude and self-control also showed the highest average score (2.36 ± 0.64), followed by Communication (2.25 ± 0.63), Fair-mindedness (2.15 ± 0.66), Fouls and body contact (1.90 ± 0.59), and Knowledge and use of the rules (1.84 ± 0.43). SOTG results between divisions showed no statistical differences, except for Communication, where U20 reveals higher scores ($p < 0.01$).

Overall SOTG mean scores are classified as good spirit (above 10 points) for both divisions, with ranges from 3 to 14 for U17 and from 3 to 16 for U20 on a 0–20 scale. Sorted by divisions (Table 2), overall SOTG mean scores are also above 10 points (Good) and statistically similar ($p \geq 0.05$) for Open, Women's, and Mixed teams (10.39 ± 2.01, 10.28 ± 1.44 and 10.18 ± 1.50, respectively).

For the U17, SOTG inferential analysis between divisions showed most categories to be very similar, except for Fouls and body contact, which had statistically significantly higher results for Open teams when compared to Women's teams ($p < 0.05$), but not when compared to Mixed teams ($p \geq 0.05$).

For the U20, inferential analysis between divisions showed most categories to be statistically similar, except for Fouls and body contact ($p < 0.05$) and Fair-Mindedness ($p < 0.05$), which had significantly higher results for Open teams when compared to Mixed teams, but not when compared to Women's teams. Overall SOTG scores were lower for the Mixed division ($p < 0.05$).

**Table 2.** Overall and detailed SOTG Scores for U17 and U20 divisions.

| U17 | Divisions | *n* | M | SD | Min. | Máx. | *p* |
|---|---|---|---|---|---|---|---|
| SOTG | Open | 80 | **10.39** | 2.01 | 3 | 14 | |
| | Women's | 46 | **10.28** | 1.44 | 6 | 14 | 0.52 |
| | Mixed | 44 | **10.18** | 1.50 | 7 | 13 | |
| Knowledge and use of the rules | Open | 80 | **1.81** | 0.62 | 0 | 3 | |
| | Women's | 46 | **1.85** | 0.42 | 1 | 3 | 0.30 |
| | Mixed | 44 | **1.70** | 0.46 | 1 | 2 | |
| Fouls and body contact | Open | 80 | **1.98** | 0.55 | 1 | 3 | |
| | Women's | 46 | **1.74** | 0.44 | 1 | 2 | 0.03 |
| | Mixed | 44 | **1.77** | 0.48 | 1 | 3 | |
| Fair-mindedness | Open | 80 | **2.10** | 0.63 | 0 | 3 | |
| | Women's | 46 | **2.11** | 0.48 | 1 | 3 | 0.49 |
| | Mixed | 44 | **2.23** | 0.52 | 1 | 3 | |
| Positive attitude and self-control | Open | 80 | **2.38** | 0.70 | 1 | 4 | |
| | Women's | 46 | **2.52** | 0.51 | 2 | 3 | 0.63 |
| | Mixed | 44 | **2.45** | 0.76 | 1 | 4 | |
| Communication | Open | 80 | **2.13** | 0.51 | 1 | 3 | |
| | Women's | 46 | **2.07** | 0.53 | 1 | 3 | 0.53 |
| | Mixed | 44 | **2.02** | 0.46 | 1 | 3 | |
| **U20** | | | | | | | |
| SOTG | Open | 108 | **10.68** | 1.74 | 5 | 15 | |
| | Women's | 92 | **10.68** | 1.70 | 4 | 15 | 0.04 |
| | Mixed | 64 | **9.92** | 2.35 | 3 | 16 | |
| Knowledge and use of the rules | Open | 108 | **1.87** | 0.39 | 1 | 3 | |
| | Women's | 92 | **1.88** | 0.33 | 1 | 2 | 0.11 |
| | Mixed | 64 | **1.73** | 0.60 | 0 | 3 | |
| Fouls and body contact | Open | 108 | **2.00** | 0.61 | 0 | 3 | |
| | Women's | 92 | **1.89** | 0.50 | 0 | 3 | 0.01 |
| | Mixed | 64 | **1.73** | 0.62 | 0 | 3 | |
| Fair-mindedness | Open | 108 | **2.20** | 0.67 | 0 | 4 | |
| | Women's | 92 | **2.23** | 0.59 | 0 | 4 | 0.04 |
| | Mixed | 64 | **1.95** | 0.72 | 0 | 3 | |
| Positive attitude and self-control | Open | 108 | **2.36** | 0.63 | 1 | 4 | |
| | Women's | 92 | **2.42** | 0.60 | 1 | 3 | 0.27 |
| | Mixed | 64 | **2.27** | 0.72 | 1 | 4 | |
| Communication | Open | 108 | **2.24** | 0.62 | 0 | 3 | |
| | Women's | 92 | **2.26** | 0.57 | 1 | 4 | 0.99 |
| | Mixed | 64 | **2.23** | 0.71 | 0 | 4 | |

Notes: U17 = under 17; U20 = under 20; M = mean; *n* = sample size; SD = standard deviation; Min = minimum; Max. = maximum.

Correlations were performed between all SOTG categories (Tables 3 and 4), and all correlate (moderately and positively) with the overall SOTG scores ($p < 0.001$), except for Fouls and body contact in the U17 division, and for Knowledge and use of rules in the U20 division, correlating with a weak intensity.

**Table 3.** Correlations between SOTG domains scores for U17 division (*n* = 170).

|  |  | 2 | 3 | 4 | 5 | 6 |
|---|---|---|---|---|---|---|
| 1 | r | 0.615 ** | 0.460 ** | 0.683 ** | 0.673 ** | 0.519 ** |
|  | *p* | 0.000 | 0.000 | 0.000 | 0.000 | 0.000 |
| 2 | r |  | 0.193 * | 0.301 ** | 0.251 ** | 0.255 ** |
|  | *p* |  | 0.012 | 0.000 | 0.001 | 0.001 |
| 3 | r |  |  | 0.221 ** | 0.036 | 0.041 |
|  | *p* |  |  | 0.004 | 0.641 | 0.593 |
| 4 | r |  |  |  | 0.410 ** | 0.168 * |
|  | *p* |  |  |  | 0.000 | 0.028 |
| 5 | r |  |  |  |  | 0.238 ** |
|  | *p* |  |  |  |  | 0.002 |

Notes: 1 = Spirit of the game; 2 = Knowledge and use of the rule; 3 = Fouls and body contact; 4 = Fair-mindedness; 5 = Positive attitude and self-control; 6 = Communication; * $p < 0.05$; ** $p < 0.01$

**Table 4.** Correlations between SOTG domains scores for the U20 division (*n* = 264).

|  |  | 2 | 3 | 4 | 5 | 6 |
|---|---|---|---|---|---|---|
| 1 | r | 0.481 ** | 0.626 ** | 0.682 ** | 0.617 ** | 0.592 ** |
|  | *p* | 0.000 | 0.000 | 0.000 | 0.000 | 0.000 |
| 2 | r |  | 0.197 ** | 0.326 ** | 0.139 * | 0.201 ** |
|  | *p* |  | 0.001 | 0.000 | 0.024 | 0.001 |
| 3 | r |  |  | 0.273 ** | 0.317 ** | 0.260 ** |
|  | *p* |  |  | 0.000 | 0.000 | 0.000 |
| 4 | r |  |  |  | 0.247 ** | 0.289 ** |
|  | *p* |  |  |  | 0.000 | 0.000 |
| 5 | r |  |  |  |  | 0.134 * |
|  | *p* |  |  |  |  | 0.029 |

Notes: 1 = Spirit of the game; 2 = Knowledge and use of the rule; 3 = Fouls and body contact; 4 = Fair-mindedness; 5 = Positive attitude and self-control; 6 = Communication; * $p < 0.05$; ** $p < 0.01$.

SOTG categories shown to be positively correlated between themselves, but with low intensity for both U17 and U20 divisions ($0.1 \leq r < 0.5$), except for Fouls and body contact, which does not correlate with Positive attitude and self-control, nor with Communication ($p \geq 0.05$).

## 4. Discussion

This article intends to familiarize the reader with the concepts of SOTG and its implementation in the sport of ultimate. A key aspect of SOTG is respect for opponents and the rules, which is known to enhance fair play [1]. The use of SOTG in ultimate in the framework of self-arbitration as a moral practice aligns well with other tools of critical pedagogy. Our experience with the scoring system has highlighted the importance of participants understanding the meaning of the results and how they may lead to a constructive reflection to improve SOTG. Additionally, scoring is more reliable if submitted immediately after the match, and with increased SC training and experience. The scoring system was designed in accordance with the expectation that teams generally display normal, good spirit.

The WFDF is a federation of 108 member associations representing flying disc sports and their athletes in more than 104 countries. WFDF is an international federation recognized by the International Olympic Committee (IOC), the International Paralympic Committee (IPC), and the International University Sports Federation (FISU). It is also a member of the Global Association of International Sports Federations (GAISF), the Association of IOC Recognized International Sports Federations (ARISF), the International World Games Association (IWGA), the International Masters Games Association (IMGA),

and the Association for the International Sport for All (TAFISA). Ultimate is considered an alternative, hybrid, non-contact sport as it contains rules, movements, and physical demands present in more common team sports such as rugby, basketball, netball, and football [39–41].

Our goals here included determining whether players of different competitive levels demonstrated similar SOTG results. The two divisions scored higher on question *Positive attitude and self-control* and lower on question *Knowledge and use of the rules.* The 2nd highest scoring question was *Fair-mindedness* (U17) and *Communication* (U20), with *Fair-mindedness* scoring similarly between U17 and U20, but that was not the case for *Communication*. The main reasons for these differences between U17 and U20 may reflect the level of maturity, levels of performance, and competitiveness between the age groups.

In the U17 division, there were two questions in which a few scores of zero values were given (*Knowledge and use of the rules* and *Fair-mindedness*). There were also instances of perfect scores of 4 (*Positive attitude and self-control*). In both cases, the SC is required to add a comment. This is not an unreasonable expectation, given that familiarity with rules, values, and traditions associated with specific sports should be sufficient for the SC to distinguish between good and bad sport practices [42].

SOTG scores for U17 and U20, by divisions (Mixed and Open) gave the lowest scores for question *Knowledge and use of the rules* and the highest scores for *Positive attitude and self-control*. In the Women's division, the lowest scoring questions were Knowledge *and use of the rules* and *Fouls and body contact*, with essentially equal scores. The need to improve the knowledge of the rules may be associated with the age of the players and their contact longevity with the sport. On the other hand, only the female division presented lower values for the question *Fouls and body contact*. As demonstrated in recent research by our group, a variety of environmental and social variables can influence the relative perceptions of athletes in different divisions, with the Master's Women's division reporting the highest ego orientation score [41].

The U20 mixed division also gave a low average score in this category. In mixed gender ultimate, discrepancies in body size and aggressiveness can create conflicts because the sport lacks a player positioning structure. Any player can cover any other in a person-to-person defensive structure, and zone defenses are common. While picks are illegal, they do happen inadvertently and can result in collisions between players with great differences in mass. There is normally no ill intent in these situations, but the player with lower mass might assume the collision was due to aggressive body contact.

We believe that this system lends itself well to adoption into other youth sports that may struggle with sportsmanship issues [43]. Similar outcomes reported from other self-arbitrated sports would be useful in further refining the system. Encouraging self-arbitration in other sports could have substantial educational value. This innovative content delivery provides inclusive lifelong learning opportunities and adds value to quality education, supporting the United Nations Sustainable Development Goals, particularly Goal number 4, Quality Education. The use of sports in this context contributes to enhancing a value-based education to further support the youth [44]. It is also aligned with the objectives of the TAFISA, as victory is not the priority, and it is accessible for all [35]. It is a limitation of our study that we were unable to obtain information on social support networks, such as player educational environments, family factors, and friendship connections, which are also fundamental for adolescent personal development. We were unable to gather data on experience levels of SC. Future studies may explore ways to understand the potential bias due to these and other demographics. Such information could assist the SD in addressing outlier scores.

## 5. Conclusions

In summary, we have examined SOTG results from JJUC 2022, which verified the intended outcomes of the score distributions. While differences between age groups and divisions were small, they are likely rooted in the diversity of the rapidly changing maturity

levels in these age spreads. The spirit elements mentioned as the most valued were "Positive attitude and self-control" and the "Fair-mindedness" in the U17 and "Positive attitude and self-control" and "Communication" in the U20 games, highlighting the fact that the SOTG results were positive in both competitions. The results also highlight the importance of the athletes' knowledge of the rules. The importance of fair play extends beyond the sports field and the rules of the game. The introduction of idealistic values of fairness through the spirit of the game will be regarded as a utopia, whereas our modern lifestyle will become the governing principle of sporting ethics.

**Author Contributions:** Conceptualization, J.P.A., L.C., H.P., F.C., E.C., W.C., Z.E., J.E.M.J. and G.E.F.; methodology, J.P.A., L.C., H.P., E.C., W.C. and J.E.M.J.; validation, J.P.A., L.C., H.P., F.C., E.C., W.C., Z.E., J.E.M.J. and G.E.F.; formal analysis, J.P.A., L.C., H.P., E.C., W.C., J.E.M.J. and G.E.F.; investigation, J.P.A., L.C., H.P., E.C., W.C., J.E.M.J. and G.E.F.; resources, J.P.A.,H.P. and J.E.M.J.; data curation, J.P.A., L.C., H.P., W.C., Z.E., J.E.M.J. and G.E.F.; writing—original draft preparation, J.P.A., L.C. and H.P; writing—review and editing, J.P.A., L.C., H.P., W.C. and J.E.M.J.; visualization, J.P.A., L.C., H.P., F.C., E.C., W.C., Z.E., J.E.M.J. and G.E.F.; funding acquisition, J.P.A., E.C. and W.C. All authors have read and agreed to the published version of the manuscript.

**Funding:** This research received no external funding.

**Institutional Review Board Statement:** The study was conducted in accordance with the Declaration of Helsinki and approved by the Chair of Ethics Committee of WFDF (8 February 2022).

**Informed Consent Statement:** Informed consent was obtained from all subjects involved in the study.

**Data Availability Statement:** Not applicable.

**Acknowledgments:** The authors acknowledge the WFDF—World Flying Disc Federation—and the EUF—European Ultimate Federation—for all their support and all the players in JJUC 2022 for their good spirit. We thank the local tournament organizers, Dulu, Dariusz, and Adam et al., and the spirit directors (Henrietta and Jakub). Many thanks to our research team for all their efforts and contributions and to SOTG Commission: Chihiro Ono (JPN), Dario Lucisano (ITA), Guo Yang (CHN), James Moore (GBR), Kate Barabanova (RUS), Kate Kingery (USA), Samir El Ajraoui (MAR), Valeriia Strelchyna (UKR), and Wolfgang Maehr (SGP). To Karina Woldt and Bruno Gravato, we offer our appreciation for support with all data. J.A. thanks the Portuguese Foundation for Science and Technology for I.P., Number (UIDB/04748/2020). G.E.F. is grateful for national support from FCT—the Foundation for Science and Technology, P.I., through the institutional scientific employment program-contract (CEECINST/00077/2021).

**Conflicts of Interest:** The authors declare no conflict of interest.

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
