# Peer review of "Self-Refereeing System in Ultimate during the Joint Junior Ultimate Championship in Three Different Divisions—A Different Way to Promote Fair-Play?"

_2673-995X, doi:10.3390/youth3010028_

Round 1

Reviewer 1 Report

It is great to see that self-refereeing in Ultimate receives more attention in academic research and this paper is a valuable contribution to this trend. Both the introduction/conclusion and the empirical study are well-conceived and clearly presented. 

My only concern is that the two aspects are not perfectly connected. A key claim of the more theoretical part is to promote Ultimate (or self-refereeing more general) as a valuable tool in pedagogic contexts. However, the empirical study is not in a position to support that claim. The same also holds for some other claims in the introduction, discussion, and conclusion. I nevertheless think that this is a valuable contribution that warrants publication. 

I noticed some typos (e.g.: “SOTG time sheet” -> “SOTG score sheet”, “If a team gives a zero on four” -> “If a team gives a zero or four”) I am sure there are more so I recommend another round of careful proofreading. 

Author Response

Dear reviewer, the comments and answers to the questions are in the attached document.

Author Response

(The authors gave the same response as above.)
